# OpenReview forum: "SafeThink: A Key to Safety in Multi-Modal Large Reasoning Models"
_ICLR.cc/2026/Conference — ICLR 2026 Conference Withdrawn Submission_

### Official Review · Reviewer_uAct · 2025-10-24

**Soundness:** 2
**Presentation:** 2
**Contribution:** 2
**Rating:** 4
**Confidence:** 2

**Summary:**

The paper proposes a prompting technique to improve safe responses via chain-of-thought under adversarial attacks in multi-model LLMs.

**Strengths:**

* The empirical evaluation of the method is comprehensive, showing significant improvement over the baselines.

**Weaknesses:**

* The novelty of the proposed approach seems limited.

* The paper claims to have “formally characterized why RL-based reasoning fine-tuning degrades safety”, however the main argument and conclusion boils down to writing a constrained optimization problem and arguing that LLMs are not trained with it. This seems a somewhat shallow conclusion to me

Minor:

> In effect, each step of reasoning is constrained to lie within the safety set $\{z : R_{\text{safe}} (x_{\text{adv}} , z) ≥ \tau\}$

Seems like a not very grounded or precise statement. The paper does not show any theoretical or empirical evidence that this is indeed the case.

Overall, while the paper does not add much in terms of new knowledge, and while the proposed method seems relatively straightforward, the strength of the empirical results outweigh these limitations in my opinion.

**Questions:**

* Why do you measure whether the model was jailbroken also in the thinking trace, even if eventually the user sees only a safe response? Can you provide your results without this evaluation?

* How statistically significant are your results? How many seeds did you use for your experiments?

---

### Official Review · Reviewer_NSvL · 2025-10-29

**Soundness:** 3
**Presentation:** 2
**Contribution:** 2
**Rating:** 4
**Confidence:** 5

**Summary:**

This paper addresses a critical vulnerability in multi-modal large reasoning models (MLRMs), where the reinforcement learning (RL) fine-tuning used to enhance reasoning capabilities inadvertently weakens safety alignment, making models more susceptible to jailbreak attacks. The authors trace this "reasoning tax" to a misspecified objective that optimizes for task accuracy while ignoring safety constraints. The primary contribution is SAFETHINK, a novel and lightweight inference-time steering method that enforces these missing constraints directly within the chain-of-thought. At each reasoning step, SAFETHINK uses a safety reward model to score the partial output; if unsafe content is detected, it intervenes by appending simple textual feedback (e.g., "Wait, think safely") to project the reasoning trajectory back into a safe state. Through comprehensive experiments on diverse safety benchmarks, the authors demonstrate that SAFETHINK significantly reduces attack success rates (e.g., by 44.57% on OpenVLThinker-7B) and reinstates safety robustness, all without retraining the model or sacrificing its reasoning accuracy on benign tasks.

**Strengths:**

1. Effective and Lightweight Method: The paper introduces SAFETHINK, a novel and practical inference-time method that effectively addresses safety vulnerabilities without costly retraining. By monitoring the chain-of-thought and using simple textual feedback to steer the model, it achieves impressive performance, significantly reducing attack success rates (e.g., by 44.57% on OpenVLThinker-7B). This approach is highly efficient as it only intervenes when unsafe reasoning is detected, thereby preserving reasoning capabilities on benign tasks.

2. Comprehensive Experimental Validation: The authors conduct a thorough and comprehensive evaluation across a diverse set of recent MLRMs, including six different state-of-the-art architectures. The method's robustness is tested against a wide array of jailbreak benchmarks, spanning both text-based attacks (JailbreakV) and multiple image-based attacks (Hades, FigStep, MM-SafetyBench). This extensive testing provides strong evidence for the generalizability and effectiveness of the SAFETHINK approach.

**Weaknesses:**

1. Limited Novelty of the Core Mechanism: The central idea of steering the model by appending textual cues shares conceptual similarities with prior work on reasoning. This approach is admittedly "inspired by" methods like "s1: Simple test-time scaling" (Muennighoff et al., 2025), which uses similar interventions (e.g., "Wait, think step by step") to improve reasoning accuracy. Therefore, the core contribution is more of a successful adaptation of this technique for safety, rather than a fundamentally new steering paradigm.

2. Unclear Justification for Multi-Modal Focus: While the paper frames the problem around multi-modal models, the SAFETHINK solution itself operates almost entirely within the text domain. The mechanism monitors textual traces and appends textual feedback, a process that seems directly applicable to text-only language models facing similar safety-reasoning trade-offs. The paper could be strengthened by either justifying the specific necessity for a multi-modal context or by demonstrating the method's applicability beyond it.

**Questions:**

n/a

---

### Official Review · Reviewer_veb6 · 2025-10-31

**Soundness:** 2
**Presentation:** 2
**Contribution:** 1
**Rating:** 2
**Confidence:** 4

**Summary:**

This paper addresses a problem known as the "Reasoning Tax" in multi-modal large reasoning models (MLRMs). This phenomenon refers to how fine-tuning a model with reinforcement learning (RL) to enhance its reasoning capabilities can paradoxically weaken its safety alignment, rendering it more susceptible to jailbreak attacks. The authors posit that this stems from a misspecified objective in RL fine-tuning, which optimizes for accuracy while neglecting safety constraints.

To address this, the paper proposes an inference-time intervention method named SAFETHINK. This method monitors "partial traces" step-by-step as the model generates its Chain-of-Thought (CoT). It employs an external safety reward model ($R_{safe}$) to evaluate the safety of each generated step. If unsafe content is detected, SAFETHINK rejects the step and injects lightweight textual feedback (e.g., "Wait, think safely.") into the context to guide subsequent generation back onto a safe trajectory. Experimental results demonstrate that this method significantly reduces the Attack Success Rate (ASR) on multiple safety benchmarks while preserving the model's reasoning capabilities.

**Strengths:**

1.  The problem this paper addresses is both real and significant. The "Reasoning Tax" phenomenon is a core challenge in the development of highly capable AI models.
2.  The experimental setup is comprehensive, covering multiple state-of-the-art MLRMs and diverse jailbreak attack benchmarks (e.g., JailbreakV, Hades, FigStep) that include both text and image modalities.
3.  The experimental results (e.g., Tables 1, 2, 3) show that the method achieves significant reductions in ASR. It also appears to maintain reasoning performance on the MathVista benchmark, outperforming several baselines.

**Weaknesses:**

1.  **Limited Novelty:** This is the most significant concern. The core mechanism of SAFETHINK is essentially applying Rejection Sampling at *each step* of the CoT. It relies on an external safety discriminator (in this case, Llama-Guard 3 8B) to monitor and score outputs, subsequently rejecting or replacing unsafe ones. This "monitor-discriminate-reject" paradigm is not novel in the safety domain.
2.  **Contribution is Dominated by the External Model:** The method's effectiveness is almost entirely dependent on the quality and capability of the external $R_{safe}$ (i.e., Llama-Guard 3). All the safety "intelligence" originates from this discriminator, while SAFETHINK merely acts as an "actuator" for it. This positions the contribution more as a clever "application" rather than a fundamental "innovation."
3.  **Reliance on $p_{safe}$:** Another critical component of the method is the injection of "Wait, think safely." While the ablation study (Table 4) shows this prompt is effective, this appears to be an empirical "prompt engineering" trick.

**Questions:**

Have the authors attempted to use other models (e.g., general-purpose LLMs, not just Llama-Guard-3-8B) for the scoring?

---

### Note · Authors · 2025-11-27

**Comment:**

Thank you very much for taking the time to review our paper. All of the reviewers' comments have helped us identify weak points for improvement in our experimental evaluation and writing. We greatly appreciate your constructive feedbacks, which have helped us understand what baselines and evaluations we need to add to improve our paper. However, we need time to improve our manuscript, and thus, we have decided to withdraw the paper.

Thank you again.

**Withdrawal Confirmation:**

I have read and agree with the venue's withdrawal policy on behalf of myself and my co-authors.